# Developmental Therapeutics in Metastatic Prostate Cancer: New Targets and New Strategies

**DOI:** 10.3390/cancers16173098

**Published:** 2024-09-06

**Authors:** Jingsong Zhang, Juskaran S. Chadha

**Affiliations:** Department of Genitourinary Oncology, Moffitt Cancer Center, Tampa, FL 33612, USA; juskaran.chadha@moffitt.org

**Keywords:** PSMA, STEAP1, KLK2, PSCA, DLL3, prostate cancer, immunotherapy, bispecific T cell engagers, CAR T, antibody–drug conjugate

## Abstract

**Simple Summary:**

The developments of radioligand therapy, immunotherapy, and targeted chemotherapy have led to dozens of clinical trials targeting tumor-associated surface antigens in metastatic prostate cancer. Emerging clinical data on the development of these new therapies were summarized and discussed.

**Abstract:**

There is an unmet need to develop new treatments for metastatic prostate cancer. With the development of targeted radioligand therapies, bispecific T cell engagers, antibody–drug conjugates and chimeric antigen receptor T cell (CAR T) therapies, tumor-associated cell surface antigens have emerged as new therapeutic targets in metastatic prostate cancer. Ongoing and completed clinical trials targeting prostate-specific membrane antigen (PSMA), six transmembrane epithelial antigens of the prostate 1 (STEAP1), kallikrein-related peptidase 2 (KLK2), prostate stem cell antigen (PSCA), and delta-like protein 3 (DLL3) in metastatic prostate cancer were reviewed. Strategies for sequential or combinational therapy were discussed.

## 1. Introduction

An estimated 299,010 new cases of prostate cancer and an estimated 35,250 deaths will occur in the United States in 2024. The estimated number of new cases and deaths was 220,800 and 27,540, respectively, in 2015. There is an urgent need to develop new treatments for this disease. Metastatic prostate cancer is not lethal until it progresses to the castration-resistant stage. Therefore, metastatic castration-resistant prostate cancer (mCRPC) has been the focus of early phase therapeutic development in prostate cancer.

Over the past 14 years, major breakthroughs have been made in targeting the androgen receptor (AR) signaling pathway. After the approval of the new generation AR signaling inhibitors (ARSIs) i.e., Abiraterone acetate (abi), Enzalutamide (enza), Apalutamide, and Darolutamide, much effort is spent to develop drugs that would overcome the cross-resistance among the ARSIs. These include the failed attempts with Galeterone [1], which inhibits CYP17 and AR, and induces AR protein degradation. An EZH2 inhibitor, CPI-1205, was evaluated in the phase 1b/II ProStar trial to overcome the cross-resistance among ARSIs. CPI-1205 failed to improve the response rate when combined with either abi or enza versus abi or enza alone in patients with mCRPC who have progressed through one ARSI (NCT03480646) [2]. Currently, oral proteolysis targeting chimera degrader of AR: ARV-110/Bavdegalutamide [3]; ARV-766 [4] and AR N terminal inhibitors EPI-7386 are under clinical development for mCRPC [5,6].

The approval of the prostate-specific membrane antigen (PSMA) PET scan for prostate cancer staging in 2020 and the approval of Lutetium Lu177 vivpivotide tetraxetan (Lu177) for treating PSMA PET-positive mCRPC in 2022 have opened up tremendous new opportunities for prostate cancer developmental therapeutics. These tumor-associated cell surface antigens such as PSMA do not need to be directly involved in prostate cancer carcinogenesis. As long as they are overexpressed in prostate cancer with low levels of expression in normal tissue, they could serve as targets for radioligand therapy. Recent advancements in cancer immunotherapy also make these tumor-associated cell surface antigens great candidates for developing bispecific T cell engagers (BiTEs), CAR T cell therapies, and antibody–drug conjugates (ADCs) [7]. In this review, we will focus on the current clinical development and treatment strategies targeting PSMA, six transmembrane epithelial antigen of the prostate 1 (STEAP1), kallikrein-related peptidase 2 (KLK2), prostate stem cell antigen (PSCA), and delta-like protein 3 (DLL3).

## 2. PSMA, STEAP1, KLK2, PSCA and DLL3

### 2.1. PSMA

Unlike the secretory protein of prostate-specific antigen (PSA), PSMA is a type II membrane protein that contains a 707-amino-acid extracellular domain, a 24-amino-acid transmembrane domain, and a 19-amino-acid intracellular domain. Prostate adenocarcinoma and its metastases are known to overexpress PSMA, which functions as glutamate-preferring carboxypeptidase. The role of PSMA in prostate cancer carcinogenesis is unclear at this moment. Expression of PSMA has also been detected in normal prostate epithelium, submandibular glands, hepatocytes, proximal tubules of the kidney, testis, and the gastrointestinal tract [8]. Although 111 Indium labeled Cyt-356/ProstaScint was approved for detecting occult prostate cancer recurrence in 1996, applying PSMA PET to stage high-risk prostate cancer was not widely adopted until after the approval of the more sensitive and more specific [68Ga] Ga-PSMA-11 in December 2020. Since then, piflufolastat F 18/Pylarify and 18F-rhPSMA-7.3/Posluma have also been approved. Other than initial staging and detecting occult metastasis for recurrent prostate cancer, these PSMA PET scans are used to select mCRPC patients for anti-PSMA radioligand therapy.

#### 2.1.1. PSMA Target Radioligand Therapy (TRT)

After the approval of Lu177 for PSMA PET-positive mCRPC in the post-chemotherapy setting [9], Lu177 and 177Lu-PNT2002 or PSMA I&T were compared to ARSI switch in mCRPC after disease progression through frontline ARSI therapy. Chemotherapy in the mCRPC setting was not allowed and a positive PSMA PET was required for enrollment. Although both are β-emitters, 177Lu-PNT2002 contains a DOTAGA chelator [10], which is different than the DOTA chelator used in Lu177. Given chemotherapy would be preferred to ARSI switch based on the CARD trial [11], both phase III studies used 2:1 randomization and the crossover design. The primary endpoint for the improvement of radiographic progression-free survival (rPFS) was met in both studies. In the second interim analysis of the PSMAfore trial, 82% of the patients who had radiographic progression in the ARSI switch arm had crossed over to Lu177. Although the primary endpoint of rPFS remained statistically significant, no statistically significant improvement in overall survival (OS), a key secondary endpoint, was observed. A trend for an OS benefit of Lu177 over ARSI switch was observed when data were analyzed according to the pre-specified crossover adjusted analysis, but not when the analysis was unadjusted [12]. To improve the efficacy of PSMA targeting radioligand therapy (TRT), ongoing early phase trials are combining Lu177 with an immune checkpoint inhibitor (ICI), enzalutamide, Poly-ADP-ribose polymerase (PARP) inhibitor, and muti-kinase inhibitor in mCRPC (Table 1) [13,14].

Radiohybrid PSMA-targeted ligands (rhPSMA) are also being developed as a novel platform for theranostic applications. 18F labeled rhPSMA-7.3/Posluma is FDA-approved as a PSMA PET tracer for stage prostate cancer. The Lu177 labeled 177Lu-rhPSMA-10.1 is being evaluated in a phase I/II study for mCRPC (NCT05413850). An intrapatient dosimetry comparison of 177Lu-rhPSMA-10.1 and 177Lu-PSMA-I&T in four mCRPC patients reported that 177Lu-rhPSMA-10.1 delivered an average of 3.3 times (range, 1.2–8.3 times) higher absorbed radiation dose to individual tumor lesions compared to 177Lu-PSMA-I&T [15]. The phase I data on four PSMA PET-positive mCRPC patients reported PSA reductions of 100, 99, 88, and 35%. All four patients received 4–6 cycles of 177Lu-rhPSMA-10.1 at 7.4–7.7 GBq per cycle. PFS was 12 months and 15 months for two patients and beyond 24 months for the other two patients [16]. 225Ac-J591, an alpha-emitting PSMA-TRT, is being tested either as monotherapy (NCT04506567) [17] or in combination with ICI, PSMA I&T, or radium 223, a bone-seeking alpha emitter in early phase trials (Table 1 and Table 2 [13]. Other TRTs under clinical development include 177Lu-DOTA-rosopatamab (TLX591), which conjugated the HuX591 monoclonal antibody with a DOTA chelator and radiolabeled with 177Lu. A phase III trial comparing TLX591 plus standard of care (SOC) versus SOC alone is ongoing on PSMA PET-positive mCRPC after progression through ARSI. Unlike the design of the PSMAfore trial, docetaxel chemotherapy is allowed in the SOC arm [18].

#### 2.1.2. PSMA BiTEs

Multiple BiTE constructs, which link PSMA to T cell costimulatory receptor CD3 or CD28, are under early phase clinical development. These include half-life extended anti-PSMAxCD3 BiTEs: Acapatamab (AMG160), TNB-585 [19], REGN4336 [20], and JNJ-081. REGN5678 is an anti-PSAMxCD28 BiTE that is being evaluated in a phase I/II trial as monotherapy or in combination with Cemiplimab [21]. LAVA-1207 is a PSMA targeting bispecific gamma-delta T cell engager and its phase 1 study in mCRPC is actively recruiting mCRPC patients (NCT05369000) [22]. The clinical developments of two earlier anti-PSMAxCD3 BiTEs, AMG340 and HPN 424, were terminated due to the less-than-expected clinical activities observed in the early phase trials [23].

The phase 1 study with JNJ-081 was completed after enrolling 39 mCRPC patients with 10 dose escalation cohorts ranging from 0.3 µg/kg to 3.0 µg/kg given intravenously (IV), and 3.0 µg/kg to 60 µg/kg given subcutaneously (SC). Transient PSA declines were noted at higher doses, but no radiographic responses were observed [24]. A total of 133 mCRPC patients were enrolled in the phase 1 study with Acapatamab. The confirmed PSA 50 response was 30.4% and the radiographic partial response was 7.4% among the 56 patients enrolled in the dose expansion phase. The median PSA progression-free survival and the median radiographic progression-free survival (rPFS) were 3.3 months and 3.7 months, respectively [25]. Cytokine release syndrome (CRS) was commonly observed at higher dose levels of PSMA targeting BiTEs and most CRS events occurred early during cycle 1 of study treatments.

**Table 2 cancers-16-03098-t002:** Summary of active trials targeting tumor-associated cell surface antigen in mCRPC or metastatic neuroendocrine prostate cancer (mNEPC).

Target	Approach	Testing Agent and Study Design	Status	NCT#
PSMA	TRT *	Escalating dose of 225Ac-PSMA-617 IV q8wk for up to 6 doses A: post-chemo post Lu177; B: post-chemo prior to Lu177, C: post Lu177, prior chemo or ARSI is not required	Ph I, recruiting	04597411
		177Lu-rhPSMA-10.1 Ph I: 7.4–7.7 GBq IV for up to 6 doses; Ph II: post-chemo and pre-chemo	Ph I/II, recruiting [16]	05413850
		177Lu-DOTA-TLX591 at 45 mCi/m^2^ × 2 doses 14 days apart plus SOC vs. SOC in post-ARSI PSMA + mCRPC; 2:1 randomization, primary endpoint rPFS	Ph III, not yet recruiting [18]	04876651
	BiTE	JANX007/anti-PSMAxCD3 activated by tumor protease 8 cohorts of IV qwk step dose escalation followed by 3 cohorts of q2wk or q3wk	Ph I, recruiting	05519449
		REGN4336/anti-PSMAxCD3, REGN4336 + Cemiplimab, REGN4336 + REGN5678/anti-PSMAxCD28	Ph I/II, recruiting	05125016
		LAVA1207/anti-PSMAxγδT cells, LAVA1207 + low-dose SC IL-2, LAVA1207 + pembrolizymab	Ph I/II, recruiting [22]	05369000
	Tri-specific	CB307/anti-PSMAxCD137xAlbumin, CB307 + pembrolizumab in PSMA + mCRPC or solid tumors	Ph I, recruiting	04839991
	CAR T	CART-PSMA-TGFβRDN, DLTs at 1–3 × 10^8^ cells	Ph I, not recruiting [26]	03089203
	ADC	ARX517/J591-AS269, dose escalation up to cohort 8 at 2.88 mg/kg IV q3wk with no DLTs. Additional cohorts with IV q3wk and q4 wk dosing is planned	Ph I, recruiting	04662580
STEAP 1	BiTE	Xaluritamig/STEAP1 XmAb 2 + 1 IV or SC, *part 1* MTD or RP2D, *part 2* post taxanes, *part 3* pre taxanes, *part 4* combine with abi or enza, *part 5*, outpatient monotherapy	Ph I, recruiting	04221542
	CAR T	STEAP1-BBζ CAR T + enzalutamide	Ph I/II, not yet recruiting	06236139
STEAP 2	CAR T	AZD0754/STEAP2-dnTGFβRII	Ph I/II, recruiting [27]	06267729
KLK2	BiTE	JNJ-78278343/anti-KLK2xCD3 IV or SC dose escalation—dose expansion	Ph I, recruiting	04898634
		JNJ-78278343/Anti-KLK2xCD3 + JNJ-87189401/Anti-KLK2xCD28	Ph I, recruiting	06095089
		JNJ-78278343/Anti-KLK2xCD3 + Cetrelimab	Ph I, recruiting	5818683
	CAR T	JNJ-75229414, dose escalation	Ph I, not recruiting	05022849
	TRT	Actinium-225-DOTA-h11B6/JNJ-69086420 dose escalation—dose expansion Dose escalation started at 50 μCi/2 mg IV q8wk	Ph I, recruiting	04644770
PSCA	CAR T	Anti-PSCA-CAR-4-1BB/TCRzeta-CD19t-expressing, no DLTs at 1 × 10^8^ cells	Ph I, not recruiting	03873805
		MSGV1-PSCA-8T28Z γδ CAR T + zoledronic acid, dose escalation—dose expansion	Ph I, recruiting	06193486
DLL3 in mNEPC	BiTE	Tarlatamab/AMG757, IV, dose escalation—dose expansion	Ph I, not recruiting	04702737
		BI764532, cohort for NEC or small cell carcinoma of any other origin	Ph I, recruiting [28]	04429087
	Tri-specific	HPN328/anti-DLL3 × CD3 × Albumin, IV qwk, q2wk, q3wk monotherapy cohort for DLL3 expressing cancers other than SCLC	Ph I/II, recruiting	04471727

Given that this field is evolving rapidly, not all ongoing studies are included in this table. * refers to Table 1 for combinational studies involving PSMA-TRT. IV: intravenous; SC: subcutaneous; DLT: dose-limiting toxicity; SCLC: small cell lung cancer; dnTGFβRII: double negative TGFβ receptor II.

JANX007 is a tumor protease-activated anti-PSMAxCD3 BiTE. A synthetic peptide mask was used to inhibit T cell engagement via CD3. This mask is linked to an albumin-binding domain to extend the half-life and a cleavable linker by tumor protease. Tumor-specific proteolysis of the cleavable linker in the tumor microenvironment separates the mask and albumin-binding domain from JANX007 and activates tumor killing [29]. The activated TCE released from the tumor is designed to be rapidly cleared from circulation. Janux recently reported its phase 1a data on 23 heavily pretreated (4+ lines of prior therapy) mCRPC patients treated with JANX007 (NCT05519449). The rates of PSA decline of 50% or above (PSA 50) were 56% among 18 patients with a first-step dose of ≥0.1 mg and 83% among 6 patients with a first-step dose of ≥0.2 mg. CRS events were only observed in patients with PSA decline and most of the CRS were grade 1/2 and occurred transiently in cycle 1. Although the sample size is small, these promising data demonstrate the potential to improve the efficacy and safety of BiTEs by selectively activating the T cells in the tumor microenvironment.

#### 2.1.3. PSMA CAR T

NCT00664196 is one of the earliest autologous anti-PSMA CAR T trials in mCRPC. A non-myeloablative conditioning chemotherapy regimen was used prior to the anti-PSMA CAR T infusion. The starting dose of CAR T in this trial was 10^9^ followed by either low-dose or high-dose IL-2 IV infusion for a month. This trial started in 2008 and was later suspended due to lack of funding. Two out of five treated patients had a PSA response and most of the treatment-emerged adverse events were attributed to IL-2 [30]. Another autologous anti-PSMA CAR T phase 1 trial conducted by Dr. Slovin et al. administered cyclophosphamide (Cy) conditioning chemo at 300 mg/m^2^ one day prior to CAR T infusion. No IL-2 was involved. For safety, the herpes simplex virus-1 thymidine kinase (hsvtk) gene is co-expressed with the P28z receptor, rendering T cells sensitive to ganciclovir for immediate T cell elimination. The preliminary data on seven mCRPC patients were reported as a meeting abstract in 2013 [31]. Among the four patients who received 1 × 10^7^ CAR T cells/kg, one had stable disease for >6 months and another had stable disease for >16 months. No responses were seen at the next dose level with 1.5–3 × 10^7^ CAR T cells/kg, and all three treated patients developed intermittent fever up to 39 °C along with increased levels of cytokines in the serum. This trial has enrolled 13 patients and is listed as active, not recruiting at clinicaltrials.gov (NCT01140373).

To enhance the efficacy of anti-PSMA CAR T, Carl June’s group developed anti-PSMA CAR T cells armored with a dominant negative TGF-β receptor (CART-PSMA-TGFβRDN) and observed enhanced cytokine secretion, resistance to T cell exhaustion, and induction of tumor eradication in the prostate cancer mouse models [32]. Dr. Hass et al. subsequently conducted a phase 1 study with autologous CART-PSMA-TGFβRDN (NCT03089203) [26]. When the data on the initial 13 mCRPC patients were published, cohort 1 (1–3 × 10^7^/m^2^) and cohort 2 (1–3 × 10^8^/m^2^) enrolled three patients each, and no lymphodepletion chemo was given prior to CAR T infusions [33]. PSA reductions of 30% or above (PSA30) were observed in only one of the three patients enrolled in cohort 1. No PSA 30 reductions were seen in cohort 2, and two out of three patients in cohort 2 developed grade 3 CRS within 12 h of infusion. Cohort 3 enrolled one patient who received lymphodepletion chemo with Cy at 300 mg/m^2^/day and fludarabine (Flu) at 30 mg/m^2^/day at days -5, -4, and -3 followed by 1–3 × 10^8^/m^2^ CART-PSMA-TGFβRDN cells infusion at day 0. Although >90% PSA reduction was observed in this patient, the patient died at day 30 from enterococcal sepsis in the setting of multimodal immunosuppression for grade 4 CRS. Six additional patients were enrolled in cohort -3 with the same lymphodepletion chemo but reduced dose of CAR T at 1–3 × 10^7^/m^2^. Two of the six patients developed PSA 30 but not PSA 50 response. No grade 3 or above treatment-related adverse events were seen in these six patients. PSMA-TGFβRDN CAR T cell engraftment was demonstrated by detecting the CAR-specific sequence in the genomic DNA from peripheral blood. This engraftment reached its peak level within 14 days and then started to decline. Such peak of engraftment is enhanced with the use of Cy/Flu lymphodepletion chemo and was the highest in the patient in cohort 3, who achieved >90% PSA reduction. Decreased serum TGFβ1 levels were also observed in this patient in cohort 3 [33].

A parallel phase 1 study with the same Cy/Flu lymphodepletion chemo followed by CART-PSMA-TGFβRDN infusion was terminated after two deaths among the 16 enrolled patients (NCT04227275). Both patients experienced DLTs at a dose level of 1–3 × 10^8^. One developed grade 5 immune effector cell associated neurotoxicity syndrome (ICANS) and macrophage activation syndrome (MAC)/hemophagocytic lymphohistiocytosis (HLH) after receiving 30% of his fractionated dose (total dose = 0.9 × 10^8^). The other grade 5 event was deemed likely related to immune toxicities based on the >100,000 ng/mL ferritin level and the elevated serum cytokine levels. More than 50% PSA decline was observed in 2/5 evaluable patients [34]. The phase 1 study with UniCAR02-T Cells and PSMA Target Module (TMpPSMA) in PSMA-positive mCRPC was recently terminated due to limited clinical activity (NCT04633148).

#### 2.1.4. PSMA ADC

As shown in Table 2, the early developments of anti-PSMA ADC in mCRPC only showed modest clinical activities. The study reported by Petrylak et al. used anti-PSMA ADC with the monomethylauristatin E (MMAE) payload [35]. Given both taxane chemotherapy and MMAE are mitosis inhibitors, the 21% PSA 50 response rate and the 53% CTC conversion rate were higher in the group of 35 chemo naïve patients compared to the post-docetaxel patients. The most common grade 3 or above treatment-related adverse events (TRAEs) were neutropenia, fatigue, electrolyte imbalance, anemia, and neuropathy. Two patients who received the 2.5 mg/kg dose died from sepsis. MLN-2704 is another anti-PSMA ADC, which uses a potent inhibitor of tubulin polymerization, DM1, as the payload. As commonly seen with other anti-tubulin chemo agents, peripheral neuropathy was the most common AE seen in the phase 1/2 trial with MLN-2704 [36]. MEDI-3726 used a DNA crosslinking agent PBD dimer as the payload [37]. Although no MTD was reached in this phase 1 study, further clinical development was stopped due to the limited anti-tumor activities (Table 3) [35,36,37,38].

Unlike the cleavable linker used in the anti-PSMA ADC mentioned above, the newest ADC, ARX 517 uses a non-cleavable linker between the antibody and its MMAF-containing payload, AS269. A non-cleavable linker requires complete lysosomal proteolytic degradation of the antibody, which improves the stability of the ADC and reduces the toxicities from the payload. To improve further the therapeutic window of ARX517, AS269 is covalently conjugated to a synthetic amino acid, para-acetyl phenylalanine (pAF), which is designed to incorporate at amino acid position 114 on the heavy chain of the anti-PSMA antibody. The anti-PSMA antibody of ARX517 is the same humanized J591 antibody as MLN2704 and MEDI3726. Preliminary data on the phase I study of ARX 517 (NCT04662580) were reported at ESMO 2023. No dose-limiting toxicities (DLTs) or serious adverse events (SAEs) were noted at the dose level of 2.88 mg/kg given by IV once every 21 days. In cohorts 6 to 8 (*n* = 23), 61, 52, and 26% of patients achieved at least 30%, 50%, and 90% PSA decline, respectively. Unlike other ADC with anti-mitosis payloads, no grade 3 or above peripheral neuropathy has been reported with ARX517 [38].

Compared to radioligand therapy, BiTE or CAR T could be more effective in treating lesions with low or heterogeneous expression of PSMA. Of note, most of the early phase trials with BiTEs and ADCs did not require prescreening for PSMA positivity. Figure 1 illustrates a case of an mCRPC patient who underwent anti-PSMA x CD28 bispecific engagers/REGN5678 followed by Lu177. This mCRPC patient had been through abi, radium 223, enza, docetaxel, and cabazitaxel before being referred to us for a phase 1 clinical trial with REGN5678 and Cemiplimab (NCT05125016). His Pylarify PSMA PET scan prior to enrollment reported avid PSMA uptakes in the widespread bone metastases but low levels of uptakes in the liver metastases. He underwent three weekly IV infusions of REGN5678 at 100 mg followed by cycle 1 of therapy with 6 weekly IV REGN5678 and 350 mg IV Cemiplimab at day 1 and day 22. Although the liver metastases had progressed on his post-cycle 1 CT scan along with rising PSA, clinically he reported improved energy and general wellbeing. Given there were no good standard of care options left at that time, he continued to study treatments based on the clinical benefits. The second study assessment scans 6 weeks later showed good partial response in the liver lesions and his PSA reduced more than 90%. He was off study treatment after the last dose of Cemiplimab at day 15 cycle 3 due to hospitalization for altered mental status. Workups were notable for elevated total protein in CSF and encephalopathy on EEG. His mental status quickly improved with IV dexamathsone 10 mg every 6 **h** and IVIG x2. He developed progressive disease in the bone 4 months after being off study drugs. He was then treated with Pluvicto and had >70% PSA decline (Figure 1b). His liver metastases remained stable before and during Lu177 treatment. Of note, Lu177 is known to have limited efficacy for liver metastases and metastases with low levels of PSMA uptakes. If this heavily pretreated mCRPC patient started with Lu177, he might not have had adequate liver function or blood counts to be eligible for the BiTE trial.

### 2.2. STEAP1

STEAP1 belongs to the STEAP family of metalloreductases that has been reported to promote cancer cell proliferation, invasion, and epithelial-to-mesenchymal transition [39,40,41,42]. It is highly expressed in >80% of mCRPC with limited expression in normal human tissue [39,43,44].

#### 2.2.1. STEAP1 ADC

The clinical developments on STEAP1 started with DSTP3086S, an ADC that contains the humanized anti-STEAP1 monoclonal antibody MSTP2109A linked to MMAE through a protease labile linker. The same MSTP2109A antibody was labeled with 89Zr-DFO for PET imaging. In a clinical study with 19 mCRPC patients, the median SUVmax was 20.6 in bone and 16.8 in soft tissue after receiving approximately 185 MBq/10 mg of 89Zr-DFO-MSTP2109A [45]. Of note, all enrolled patients are required to have STEAP1 1+ or above positivity on tumor tissue immunohistochemistry (IHC). Unlike the current FDA-approved PSMA PET tracers, low levels of kidney uptakes were noted with 89Zr-DFO-MSTP2109A, and 16 of the 17 PET-positive lesions were biopsy proven to be prostate cancer.

In the phase I study with DSTP3086S, a total of 84 mCRPC patients with positive STEP1 IHC were enrolled. Grade 3 transaminitis, grade 3 hyperglycemia, and grade 4 hypophosphatemia were the DLTs observed at the dose escalation phase with 35 patients. The dose expansion cohort started with 2.8 mg/kg once every 3 weeks in the initial 10 patients but was reduced to 2.4 mg/kg once every 3 weeks in the other 39 patients due to frequent dose reductions [46]. Among 62 patients who received >2 mg/kg DSTP3086S once every 3 weeks, the PSA50 response was 18%, and 59% of patients had CTC conversion from ≥5 to <5/7.5 mL of blood. 89Zr-DFO-MSTP2109A PET/CT was performed as an exploratory endpoint for this trial. There was no correlation between SUVmax tumor uptake and STEAP1 IHC, survival after ADC treatment, number of ADC treatment cycles, or change in PSA level [45].

#### 2.2.2. STEAP1 BiTEs

AMG 509, or Xaluritamig, is a first-in-class STEAP1 XmAb 2+1 T cell engager. It contains two Fragment antigen binding (Fab) domains that bind STEAP1, a CD3ε-binding single chain Fragment variable (scFv) domain, and a Fragment crystallizable (Fc) immunoglobulin domain. CD3ε is a subunit of the T cell receptor. Preclinical studies have shown that AMG 509 induced potent T cell-dependent cytotoxicity in vitro and prostate cancer regression in vivo. It has minimal cytotoxicity in normal cell lines and the AR-negative PC3 prostate cancer line [44].

The phase 1 study of AMG 509 (NCT04221542) was started in 2020 and planned to enroll 461 patients with mCRPC. Part 1 of this trial evaluates AMG509 given by IV in mCRPC patients previously treated with ARSI and up to two prior taxanes. Part 2 evaluates SC dosing of AMG 509 in the same patient population. Part 3 will explore AMG 509 IV dosing in chemotherapy-naïve patients previously treated with one ARSI, and part 4 will evaluate the combination of AMG 509 with abiraterone (4A) or enzalutamide (4B) in patients previously treated with 1 ARSI and up to one taxane (Table 3).

Clinical results from the part 1 dose exploration cohort were reported in October 2023 and published in January 2024 [47]. Ninety-seven patients received >1 IV dose ranging from 0.001 to 2.0 mg weekly or every 2 weeks. A step-dosing strategy was used for dose escalation. MTD started with a priming dose of 0.1 mg, followed by a day 8 dose of 0.3 mg, a day 15 dose of 1.0 mg, and then 1.5 mg IV weekly from day 22. The most common TRAEs were CRS (72%), fatigue (45%), myalgias (34%), and pyrexia (32%). Similar to other BiTE studies, CRS primarily occurred at cycle 1 with the highest grade of grade 3, which occurred on two occasions (2%). Grade 3 TRAEs occurred in 55% of patients, with only anemia (13%), myalgia (12%), and fatigue (11%) being reported in ≥10% of patients. In the PSA evaluable set of 87 patients, the RECIST objective response rate (ORR) across all doses was 24%, with a PSA50 of 49%. Greater responses were seen at target doses >0.75 mg with 41% ORR and PSA50 of 59%. More importantly, 13 of 52 (25%) patients in the high-dose cohorts were on treatment for more than 6 months at the data cutoff. These are probably the most promising efficacy and safety data on BiTE therapy in mCRPC reported to date.

#### 2.2.3. STEAP1 CAR T

Preclinical work with a human STEAP1 knock-in mouse model demonstrated STEAP1 CAR T activity even in a low antigen density setting [48]. This work has led to the development of STEAP1-BBζ CAR T cells and the subsequent phase I/II evaluating STEAP1 CAR T in combination with enzalutamide in mCRPC (NCT06236139). Enrolled patients will undergo leukapheresis followed by conditioning chemotherapy with IV Cy-Flu at days -5, -4, and -3. STEAP1 CAR T will be given on day 0 along with daily enzalutamide starting at day 0. The patient will also undergo tumor biopsy at baseline, at day 14, and optionally at progression. The estimated enrollment is 48 patients.

### 2.3. KLK2

KLK2 belongs to the highly conserved serine proteases of the KLK family [49]. A biomarker study from Bonk et al. reported 78% shared sequence homology to PSA. IHC on 9576 prostate cancers noted KLK2 was negative in 23%, weak in 38%, moderate in 35%, and strong in 4% of prostate cancers [50]. A more recent study reported that strong KLK2 IHC staining was associated with short metastasis-free survival and prostate cancer-specific survival [51]. A preclinical study also indicates an oncogenic role of KLK2 in the process of extracellular matrix degradation, local invasion, cancer cell proliferation, and angiogenesis [52]. KLK2 may therefore represent a novel therapeutic target as well as a biomarker for prostate cancer [53].

#### 2.3.1. KLK2 BiTEs

JNJ-78278343 is a lead BiTE under clinical development to target KLK2. It has the standard BiTE design with one arm binding to the CD3 receptor and the other arm binding to KLK2 surface antigen. The phase 1 study of JNJ-78278343 is currently enrolling mCRPC patients in the US and Europe (NCT04898634). Both IV and SC injections of JNJ-78278343 will be evaluated in the dose escalation phase. In the dose expansion phase, participants will receive JNJ-78278343 either SC or IV at RP2D. Another ongoing phase 1 study tests the combination of JNJ-78278343 with JNJ-8718904, an anti-PSMAxCD28 BiTE (NCT06095089). A third phase 1 study evaluating the combination of JNJ-78278343 with anti-PD1/Cetrelimab is also ongoing with an estimated enrollment of 20 patients (NCT05818683).

#### 2.3.2. KLK2 CAR T

To obtain a full assessment of the therapeutic potential of KLK2, Janssen Research & Development also developed JNJ-75229414, a genetically modified CAR T targeting KLK2. The phase 1 study with JNJ-752291414 has closed enrollment after enrolling 15 patients (NCT05022849). No results have been released yet.

#### 2.3.3. KLK2 TRT

Hu11B6 is a humanized IgG1 antibody that can get internalized to prostate cancer cells after binding to the catalytic cleft of human KLK2. Preclinical studies have reported the anti-tumor activities of hu11B6 labeled with Lutetium-177 and Actinium-225 [54,55]. Actinium-225-DOTA-h11B6 or JNJ-69086420 is currently under phase 1 development in mCRPC (NCT04644770). Unlike the beta particle used in Lu177, Actinum 225 is a high-energy alpha particle. To limit the potential toxicities, patients who have received Radium-223, another high-energy alpha particle, are excluded from this study. The dose expansion part of this, phase I, will have three cohorts: pre-taxane, post-taxane, and post-Lu177. The starting dose of Actinium-225-DOTA-h11B6 in the dose escalation part was 50 μCi/2 mg given by IV once or multiple doses given IV every 8 weeks [56].

### 2.4. PSCA and PSCA CAR T

PSCA is a glycosylphosphatidyl inositol-anchored cell surface protein. The expression of PSCA is low in normal epithelial cells and is overexpressed in over 80% of prostate cancers and further enriched in prostate cancer bone metastases. Increased expression of PSCA is reported to be correlated with a higher Gleason score, higher tumor stage, grade, and castration-resistant stage [57,58]. Over-expression of PSCA has also been reported in pancreatic adenocarcinoma.

Most of the clinical developments on PSCA are early phase CAR T trials. Dorff et al. recently published their phase 1 data on PSCA-targeted 4-1BB-co-stimulated CAR T (NCT03873805) [59]. All 14 enrolled mCRPC patients received 100 million CAR T cells. Three did not receive LD chemo, six received the standard Cy-Flu regimen at 500 mg/m^2^ and 30 mg/m^2^ for 3 days, and the other five had Cy-Flu at 300 mg/m^2^ and 30 mg/m^2^ for 3 days after 2/6 patients developed dose-limiting cystitis with Cy at 500 mg/m^2^. A total of 79% of screened patients had PSCA expression 2+ or above in more than 80% of cancer cells. Consistent with the PSMA CAR T study, CAR T expansion was greater with LD chemo. All three patients without LD chemo had progressive disease. Seven out of eleven patients who received LD chemo had stable disease, including one patient with reduced metastases burden in the liver. One out of five of the patients in the reduced Cy cohort had PSA50 response. At the 100 million cells, this anti-PSCA CAR T is deemed safe with no ICAN, MAS or grade 3 or above CRS observed.

BPX-601 is an autologous PSCA targeting CAR T with a rimiducid-inducible MyD88/CD40 co-stimulation switch to enhance T cell potency and persistence. The preliminary results on eight mCRPC patients enrolled in the phase 1 study of BPX-601 (NCT02744287) were reported at the ASCO GU symposium in February 2023. After LD chemo, all eight patients received a single dose of BPX-601 CAR T cells at 5 × 10^6^/kg. This was followed by a single (three patients) or weekly (five patients) dose of rimiducid at 0.4 mg/kg started 7 days after CAR T infusion. PSA 50 response was noted in three patients on day 28. One PR and three SD were also noted, including one SD for >9 months. Rapid increases in INF-γ, GM-CSF, and IL-6 were noted 24 h after rimiducid treatment. Serious AEs include two grade 3 CRS, one grade 4 ICAN, and one grade 5 neutropenic sepsis with possible HLH. The clinical development of BPX-601 was discontinued in March 2023 after another DLT with grade 4 CRS.

Unlike the alpha beta T cells used in most solid tumor CAR T trials, gamma delta (γδ) T cells represent an attractive vehicle for CAR T cell therapy given their enhanced cytotoxicity and safety. The γδ T cells could also synergize with zoledronic acid in the bone-cancer microenvironment due to their ability to recognize phosphoantigens such as isopentenyl pyroposphate (IPP) [60,61,62]. Abate-Daga et al. recently reported their preclinical work on the activity of γδ enriched CAR T cells targeting PSCA in their murine mCRPC models. Of note, administration of zoledronic acid resulted in CAR-independent activation of γδ CAR T cells, increased cytokine secretion, and enhanced anti-tumor activities in vivo [63]. This has led to the first γδ enriched autologous CAR T trial in prostate cancer (NCT06193486). This phase 1 study is enrolling mCRPC patients at the Moffitt Cancer Center. Infusion of zoledronic acid is required prior to leukapheresis at day-14, which will be followed by the standard Cy-Flu lymphodepletion chemo at days-5, -4, and -3. Fresh γδ enriched CAR T cells targeting PSCA will be infused at day 0 (Figure 2). Based on a recent news release, the first patient with heavily pretreated mCRPC has been successfully dosed at the starting dose at 1.5 × 10^5^ cells/kg.

### 2.5. DLL3

Transformation of prostate adenocarcinoma to small cell neuroendocrine carcinoma is characterized by an aggressive clinical course, and poor prognosis, with limited therapeutic options. Expression of PSMA is known to be lost in neuroendocrine prostate cancer (NEPC). Given that the expression of KLK2 is also regulated by AR, the KLK2 level will likely be diminished in NEPC as well. Notably, DLL3, an inhibitory Notch ligand, has shown aberrant expression on the surface of small cell lung cancer (SCLC) and high-grade neuroendocrine tumors or small cell cancer outside the lung. The IHC study by Puca et al. reported 36/47 (76.6%) NEPC and 7/56 (12.5%) mCRPC expressed DLL3. Minimal to no expression of DLL3 was noted in localized prostate cancer (1/194) and benign prostate (0/103). Furthermore, the transcription of DLL3 is regulated by achaete-scute family bHLH transcription factor 1/ASCL1. ASCL1 inhibits the Notch singling by transactivating DLL3. Treatment of DLL3-expressing prostate cancer xenografts with a single dose of anti-DLL3 ADC, SC16LD6.5, resulted in complete and durable responses [64].

#### 2.5.1. DLL3 ADCs

Rovalpituzumab tesirine (Rova-T) or SC16LD6.5, an ADC targeting DLL3, has been studied in multiple clinical trials. There was early activity seen in preclinical models and early phase trials in DLL3 expressing advanced solid tumors as well as recurrent SCLC [65,66,67]. Rova-T versus Topotecan was subsequently tested as a second-line systemic therapy for advanced SCLC with high levels of DLL3 expression in the phase 3 TAHOE trial. This study was terminated early due to inferior OS and higher rates of serosal effusions, photosensitivity reaction, and peripheral edema in the Rova-T arm [68]. Rova-T was also compared with placebo as a maintenance therapy after frontline platinum-based chemotherapy in advanced SCLC in the phase III MERU study. This trial was also terminated early due to inferior OS [69].

#### 2.5.2. DLL3 BiTE

Tarlatamab, or AMG757, is an anti-DLL3x CD3 BiTE. Its phase II trial on 220 previously treated SCLC patients reported 40% and 32% response rates in patients in the 10 mg group and the 100 mg group, respectively. Both groups received IV Tarlatamab given once every 2 weeks. The duration of response was at least 6 months in 40 of the 68 responders. CRS is the most common AE and grade 3 CRS was 1% in the 10 mg group and 6% in the 100 mg group. Only 3% of patients discontinued Tarlatamab due to TRAEs [70]. Based on these promising phase 2 data, Tarlatamab received FDA accelerated approval for extensive-stage SCLC who have relapsed after platinum-based first-line chemotherapy on 16 May 2024. The confirmatory phase 3 randomized trial comparing Tarlatamab with DNA topoisomerase inhibitor for extensive-stage SCLC patients after platinum-based chemotherapy is ongoing (NCT05740566). The initial data of phase 1b study of Tarlatamab in patients with de novo metastatic or treatment-emergent NEPC (NCT04702737) were presented at the ASCO 2024 meeting [71]. A total of 40 NEPC patients received at least one dose of Tarlatamab and 18 of 32 biopsy evaluable patients had 1% or above tumor DLL3 positivity. Four of the eighteen DLL3 positive patients had objective responses based on RECIST 1.1 with the duration of response of 25.8 months, 9.2 months, 5.5 months, and 3.7 months, respectively. Most CRS occurred during cycle 1 with only one grade 3 CRS noted.

Unlike Tarlatamab, HPN328 is a tri-specific T cell engager with three binding domains: anti-DLL3 for target engagement, anti-albumin for half-life extension, and anti-CD3 for T cell engagement and activation. HPN328 is currently being evaluated in a phase 1/2 study as monotherapy or in combination with atezolizumab in patients with advanced, pretreated malignancies that expressed DLL3 (NCT04471727). Interim results of 66 patients who received HPN328 monotherapy from 0.015 to 24 mg across 14 dose escalation cohorts were presented at the 2024 ASCO GU symposium in January 2024. Of the 66 patients reported, 10 were NEPC and 2 were small cell bladder. Three of six imaging evaluable NEPC patients had unconfirmed PR. Both small cell bladder cancer patients had confirmed PR. Given that CRS from BiTE therapy tends to occur early, each enrolled patient started with a priming dose before being treated with the target testing dose to improve safety. Nearly all CRS occurred with the initial priming dose. Two grade 3 CRS occurred with a priming dose of 2 mg. Target dose escalation was continued to 24 mg with a priming dose of 1 mg with no further DLTs [72].

BI764532 is another anti-DLL3x CD3 BiTE that is undergoing early phase development for DLL3 expressing SCLC and neuroendocrine tumors (NEC) (NCT04429087) [28]. This study tested three IV dosing strategies: weekly, every 3 weeks, and step-in/priming followed by a fixed target dose. When the initial results on 70 patients were reported at the ASCO annual meeting in 2023, MTD was not reached. The ORR in patients who received the target dose was 33% in SCLC (*n* = 24) and 22% in NEC (*n* = 23) [73]. In December 2023, BI764532 received FDA fast-track designation for DLL3 + large-cell NEC of the lung.

## 3. Conclusions

As summarized in Table 3, the developmental therapeutics for targeting tumor-associated cell surface antigens have expanded rapidly in the past 5 years [16,18,22,26,27,28]. The recent accelerated approval of Tarlatamab for extensive-stage SCLC represents a major breakthrough in treating this lethal disease and highlights the therapeutic potential of BiTE therapy in solid tumors. Although the tumor microenvironment of mCRPC has been shown to be enriched with TGFβ, T regulatory cells, and myeloid-derived suppressor cells that would suppress the anti-tumor immune response [7], the promising efficacy and safety data from part 1 of the phase 1 study with Xaluritamig monotherapy indicate that it is feasible to engage T cells in the immunosuppressive microenvironment of mCRPC with BiTE therapy. Compared to CAR T, BiTE therapies could work better in solid tumors due to their repetitive dosing and relatively good safety profile. Of note, no lymphodepletion chemo is required for BiTE. Most BiTE-related CRS occurred early during cycle 1 and most of the AEs can be managed in the outpatient setting. The efficacy and safety of BiTE therapy can be improved further with the development of the next-generation tumor protease-activated BiTE such as JANX 007.

Early attempts with CAR T cell therapy trials in prostate cancer have shown the importance of lymphodepletion chemo for CAR T engraftment. Although several CAR T trials in mCRPC have terminated early due to grade 5 AEs and or modest responses, ongoing studies are looking to improve the safety and efficacy with gamma delta T cells enriched CAR T against PSCA (NCT06193486) and to target STEAP2 with double negative TGFβ receptor II armored CAR T (NCT06267729) [27].

The responses of mCRPC to ADCs have been modest so far. Part of the reason is due to the commonly used anti-microtubule ADC payloads such as MMAE and MMAF share a similar mechanism of action with docetaxel and cabazitaxel. To overcome the potential cross-resistance and to improve the stability of ADC, the next-generation ADC such as ARX 517 uses a non-cleavable linker between the antibody and payload. Its MMAF-containing payload AS269 is non-cell-permeable and is conjugated to a synthetic amino acid pAF, which is biosynthetically incorporated into the amino acid 114 of the heavy chain of the anti-PSMA antibody. ARX517 has demonstrated a promising efficacy and safety profile based on the phase 1 data released so far. The pros and cons of different strategies in targeting tumor surface antigens in mCRPC are summarized in Table 4.

These exciting therapeutic developments in targeting PSMA, KLK2, STEAP1, and PSCA in mCRPC have opened tremendous opportunities for sequencing treatment targeting the same tumor surface antigens with different strategies (Figure 1) or different tumor surface antigens with the same or different strategies. Early phase treatment combination trials are already ongoing to combine ICI or ARSI with TRT and BiTE therapies (Table 1 and Table 3). Combining BiTE therapies with different designs could also improve the efficacy by engaging T cells to target two surface antigens with both the CD3 and CD28 antibodies (NCT 06095089 and NCT05125016). Furthermore, antibodies that were used for TRT or ADC development for KLK2 and STEAP1 can be labeled with radionuclides for PET tracer development, which, along with the PSMA PET, can be used to identify the right patient for the right treatment at the right moment.

## Figures and Tables

**Figure 1 cancers-16-03098-f001:**
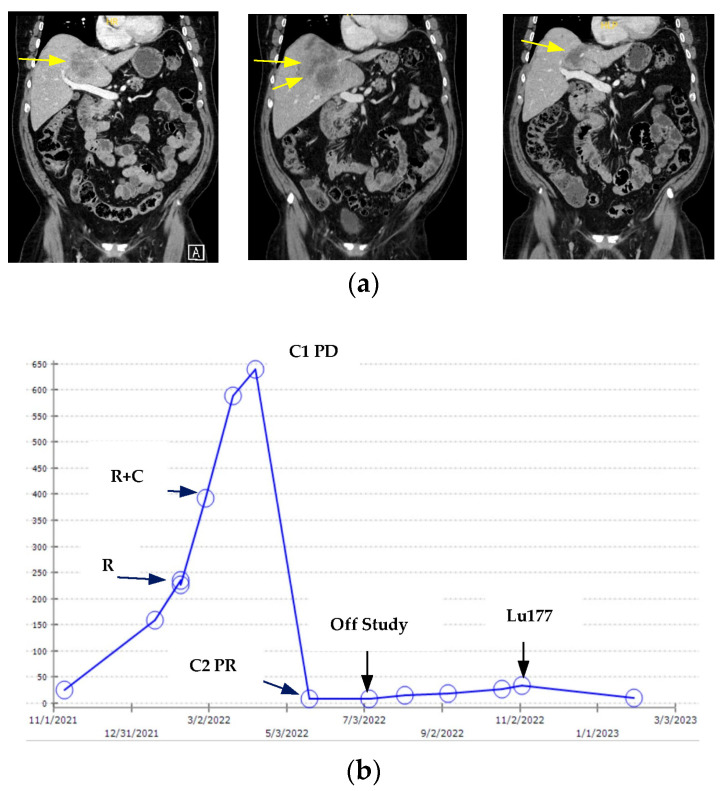
Clinical course of an mCRPC patient who was enrolled in the anti-PSMAxCD28 BiTE study followed by Lutetium Lu177 vivpivotide tetraxetan (Lu177). (**a**) Coronal CT images of baseline (**left**), post cycle 1 (**middle**), and post cycle 2 (**right**) liver metastases (yellow arrow) before and during treatment with anti-PSMAxCD28/REGN5678 and Cemiplimab per NCT05125016 protocol. (**b**) Changes in PSA values (y-axis) over time. R: REGN5678; C: Cemiplimab; C1PD: post cycle 1 progressive disease in the liver; C2PR: post cycle 2 partial response in liver.

**Figure 2 cancers-16-03098-f002:**
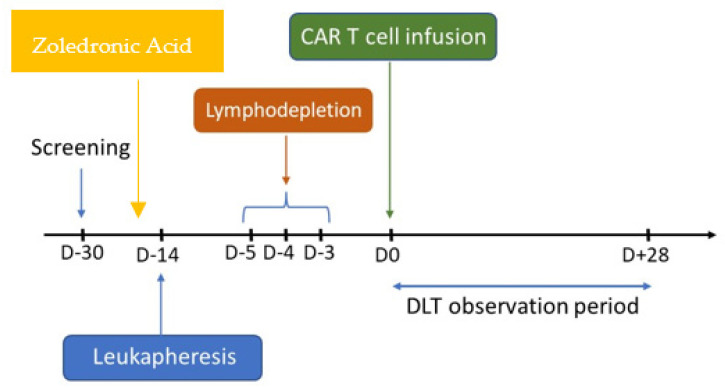
Schema of the phase I γδ enriched autologous CAR T trial targeting PSCA in mCRPC (NCT06193486). Zoledronic acid will be given prior to day-14 leukapheresis. The lymphodepletion regimen includes cyclophosphamide 500 mg/m^2^ and fludarabine (30 mg/m^2^) administered over 3 days (days-5, -4, -3). Fresh CAR T product is infused at day 0, which will be followed by a 28-day observation period for dose-limiting toxicity (DLT).

**Table 1 cancers-16-03098-t001:** Representative ongoing early phase combined treatment trials involving PSMA radioligand therapy in mCRPC.

Intervention	Phase/ Accrual Goal	NCT#	Primary Endpoint
Lu177 ^1^ + Ipi ^2^ 3 mg/kg Q 6 wk × 4 Nivo ^3^ 1 mg/kg Q 3 wk × 4 → Nivo Q 4 wk × 18	II/100	05150236	12 Month PSA-PFS ^4^
Lu177 × 1→Pembro ^5^ + Lu177 × 1 at PSA progression→ repeat Lu177 + Pembro at the next post treatment PSA progression	II/48	05766371	12 Month rPFS
177Lu-PSMA-I&T 6.8 GBq Q 8 wk + escalating dose of 225Ac-J591	I/II/48	04886986	DLT ^6^, MTD ^7^, PSA 50 RR ^8^ [13]
177Lu-PSMA-I&T + Radium-223	II/36	05383079	DLT, MTD, 50% PSA RR
Lu177 + Olaparib	I/52	03874884	DLT, MTD, RP2D ^9^
Lu177 + escalating dose of Cabazitaxel	I/II/44	05340374	DLT, MTD, RP2D [14]
Lu177 + escalating dose of Cabozantinib	I/33	05613894	MTD, PFS at 24 weeks
Lu177 + Enzalutamide vs. Enzalutamide	II/162	04419402	PSA-PFS

^1^ Lutetium Lu177 vivpivotide tetraxetan (Lu177), ^2^ Ipilimumab (Ipi), ^3^ Nivolumab (nivo), ^4^ radiographic progression-free survival (rPFS), ^5^ Pembrolizumab (Pembro), ^6^ dose-limiting toxicity (DLT), ^7^ maximum tolerated dose (MTD), ^8^ response rate (RR), ^9^ recommended phase 2 dose (RP2D).

**Table 3 cancers-16-03098-t003:** PSMA antibody–drug conjugates in mCRPC.

Drug	Antibody	Payload	Dosing	Patient #	PSA 50 Response	Best Imaging Response
PSMA-MMAE	IgG1	MMAE	2.3 or 2.5 mg/kg IV once q 21 d	119	14%	2% PR, 63% SD [35]
MLN2704	J591	DM1	330 mg/m^2^ q 2–6 wk	62	8%	No Data [36]
MEDI3726	J591	PBD dimer	0.2–0.3 mg/kg IV once q 21 d	33	3%	3% PR, 36% SD [37]
ARX517	J591	AS269	2.88 mg/kg IV once q 21 d	23 *	52%	2/9 ** PR, 4/9 SD [38]

* PSA response was reported for 23 patients treated with 2.0 mg/kg, 2.4 mg/kg, or 2.88 mg/kg ARX 517 from cohorts 6–8. ** 9 pts from cohorts 4 (1.4 mg/kg)—8 had measurable lesions and were included for response assessment per RECIST 1.1.

**Table 4 cancers-16-03098-t004:** Comparison of different strategies in targeting tumor surface antigens in mCRPC.

	Pros	Cons
ADC	Improved efficacy and safety compared to traditional chemotherapy. Potential bystander effect to mCRPC with no or low levels of antigen expression.	Cross-resistance between taxanes and anti-microtubule payloads. Limited potential for treatment combinations.
BiTEs	Documented efficacy in early phase trials. Early onset of CRS. CRS and ICAN can be mitigated with stepwise dosing. No bone marrow suppression. Potential to be combined with another BiTE, ICI, and ARSI.	Frequent IV or SC dosing. Side effects such as CRS often require inpatient overnight observation at the initial dosing.
CAR T	Potential for a deep and durable response after one infusion.	Requirement for conditioning chemo prior to CAR T infusion. Potentially life-threatening CRS, ICANS, and HLH, require inpatient observations and management.
TRT	Radiotracers can be added to the same chemical backbone for diagnosis, treatment selection, and treatment. Easy to administer. More manageable toxicities. Potential to be combined with ARSI, PARPi, and ICI.	Fixed dosing regardless of disease burden. Limited activities for metastatic lesions with low levels of surface antigen expression. Bone marrow suppression, which is more notable with alpha particles and TRT with long half-life.

CRS: cytokine release syndrome; ICANs: immune effector cell associated neurotoxicity syndrome; HLH: hemophagocytic lymphohistiocytosis; ARSI: androgen receptor signaling inhibitor; PARPi: Poly (ADP-ribose) polymerase inhibitor; ICI: immune checkpoint inhibitor.

## Data Availability

Making original data available is not applicable to this review.

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
