# Peer review of "Developmental Therapeutics in Metastatic Prostate Cancer: New Targets and New Strategies"

_cancers, 2024, doi:10.3390/cancers16173098_

Round 1
Reviewer 1 Report
Comments and Suggestions for Authors This is a reasonable review of the subject and acceptance is recommended.Author Response
There are no reviewer comments to address.
Reviewer 2 Report
Comments and Suggestions for Authors
The authors summarized and discussed new therapy in prostate cancer based on targeting on PSMA, STEAP1, KLK2, PSCA, DLL3, which is a interesting and beautiful review paper. Here are some questions as follow:
1. Please add a Briefly introduction for STEAP1
2. DDL3 and STEAP1 also have ADC in clinical trial. While it in Table 3. This info is missing. Please add it.
3. It is interesting that KLK2 can be used as a BiTE and CAR-T target, since it is a secret protein and some studies also show loss of KLK2 is in the poor prognosis prostate cancer [PMID: 32628300]. Can authors explain what is the mechanism of using KLK2 to anchor immune cells with prostate cancer cells?
4. There is some small should be improved, such as in the line 22, STEAAP1 is a typo. In Line 166, line 168, 1 x107 should be wrote as 1 x107, or 1 x10^7
Author Response
- Please add a Briefly introduction for STEAP1
Thanks for pointing this out. This is added.
- DDL3 and STEAP1 also have ADC in clinical trial. While it in Table 3. This info is missing. Please add it.
DLL3 ADC was discussed in section 2.5.1 of the original draft.
STEAP1 ADC is added to the manuscript.
- It is interesting that KLK2 can be used as a BiTE and CAR-T target, since it is a secret protein and some studies also show loss of KLK2 is in the poor prognosis prostate cancer [PMID: 32628300]. Can authors explain what is the mechanism of using KLK2 to anchor immune cells with prostate cancer cells?
We have noted the discrepancy on KLK2 in the literature. Part of it is attributed to the quality of the anti-KLK2 antibodies used in biomarker research. Based on the published data and data presented at the June 2024 ASCO meeting, prostate cancer cells do express KLK2 on its surface. KLK2 PET scan is also under development for metastatic prostate cancer. The exact mechanism on using KLK2 to anchor immune cells with prostate cancer cells is not published.
- There is some small should be improved, such as in the line 22, STEAAP1 is a typo. In Line 166, line 168, 1 x107 should be wrote as 1 x107, or 1 x10^7
We have corrected these typos.
Reviewer 3 Report
Comments and Suggestions for Authors
This review manuscript is well-organized and clearly articulated. It provides an effective summary of the development and treatment strategies aimed at PSMA, STEAP1, KLK2, PSCA, and DLL3. I believe this is a timely and comprehensive overview of an important clinical and scientific topic.
However, there is a missing second subheading, and a rearrangement of the manuscript that eliminates some study descriptions would enhance the review.
Author Response
Comment "However, there is a missing second subheading, and a rearrangement of the manuscript that eliminates some study descriptions would enhance the review."
Thanks for reviewing our manuscript. We have made changes as suggested.